# Versatile Covalent Postsynthetic Modification of Metal Organic Frameworks via Thermal Condensation for Fluoride Sensing in Waters

**DOI:** 10.3390/bioengineering8120196

**Published:** 2021-11-29

**Authors:** Eugenio Hernan Otal, Manuela Leticia Kim, Yoshiyuki Hattori, Yu Kitazawa, Juan Paulo Hinestroza, Mutsumi Kimura

**Affiliations:** 1Department of Chemistry and Materials, Faculty of Textile Science and Technology, Shinshu University, Ueda 386-8567, Japan; manuela_kim@shinshu-u.ac.jp (M.L.K.); hattoriy@shinshu-u.ac.jp (Y.H.); yu_kitazawa0311@shinshu-u.ac.jp (Y.K.); 2COI Aqua-Innovation Center, Shinshu University, Ueda 386-8567, Japan; 3Research Initiative for Supra-Materials, Shinshu University, Ueda 386-8567, Japan; 4Department of Fiber Science and Apparel Design, Cornell University, Ithaca, NY 14853, USA; jh433@cornell.edu

**Keywords:** fluoride, sensors, drinking water, metal organic frameworks, post-functionalization, thermal condensation

## Abstract

Having access to safe drinking water is one of the 17 sustainable development goals defined by the United Nations (UN). However, many settlements around the globe have limited access to drinkable water due to non-anthropogenic pollution of the water sources. One of those pollutants is fluoride, which can induce major health problems. In this manuscript, we report on a post synthetic functionalization of metal organic frameworks for the sensing of fluoride in water. The proposed thermal condensation methodology allows for a high yield of functionalization using few steps, reducing reagent costs and generating minimal by-products. We identified a Rhodamine B functionalized Al-BDC-NH${}_{2}$ metal organic framework as one particularly suitable for fluoride detection in water.

## 1. Introduction

Fluoride is an element of high interest for human health. Moderate concentrations (1 to 1.5 mg·L${}^{-1}$) are recommended to strengthen the dental enamel, but higher concentrations can cause dental fluorosis (1.5 to 3 mg·L${}^{-1}$), mottling, browning, severe teeth deterioration, and skeletal fluorosis (4–8 mg${}^{-1}$). Fluoride has no color, odor or taste, and high levels of fluoride in natural waters are mainly due to geological composition of riverbeds. This generates an inhomogeneous distribution of fluoride contents in the water sources of a certain area. The provision of a portable sensor to the local communities will help to discriminate the safe and unsafe water sources until the construction of a safe water network is constructed. One of the main areas on the planet affected by high concentrations of fluoride is the East African Rift Valley, where the concentrations of fluoride can reach up to 75 mg·L${}^{-1}$ [1], exceeding the recommended highest concentration of fluoride in drinking water by the World Health Organization of 1.5 mg·L${}^{-1}$ [2]. Access to safe drinking water is a goal of the 17 sustainable development goals defined by the United Nations (UN) [3].

The importance of reliable, cost effective and easy to use of fluoride detection systems assuring clean and safe drinkable water is highlighted [4]. Among new materials capable of detecting fluoride in natural waters are the Metal organic Frameworks also known as MOFs [5], which are a family of coordination polymers. In this multi-faceted family of compounds, the change in one of their components can trigger deep changes in their functionality. For example, by changing the length of the ligand, the isoreticular chemistry can provide compounds with the same metal–ligand connectivity and crystallographic space group, but with different cell parameters. Similarly, by using post-synthetic modifications, one can keep the same structure and add functionalities in the same crystallographic structure. These many possibilities allow the construction of “The Garden of Forking Paths” [6] where scientists of different branches of science cross ways. Additionally, the contribution of chemists to keep “The Garden of Forking Paths” green and leafy is also present [7,8,9,10].

Post-synthetic modification (PSM) is the simplest path to give MOFs some extra functionalities while preserving their unique characteristics, like crystallinity, structure, and porosity. Even though there are previous reports on PSM dating back to 1999 [11], the PSM concept was settled in 2007 [12,13], and since that moment, the chemical toolbox for MOF post-synthetic modification has grown significantly [14,15,16,17,18,19,20]. At first glance, PSM techniques can be classified into coordinative PSM and covalent PSM, which can also be sub-divided into post-synthetic deprotection (PSD), post-synthetic exchange (PSE), post-synthetic insertion (PSI), and post-synthetic polymerization (PSP) [15]. Coordinative PSM requires of an open metal site and covalent PSM requires of a synthon group in the ligand. In the case of coordinative PSM, a coordination position in the metal center should be bonded to a non-structural ligand to facilitate the removal of the ligand.

For covalent PSM, several synthons attached to the aromatic ring have been used, e.g., -CHO [21], -NH${}_{2}$ [22], and -CH${}_{3}$ [23], among others. The more versatile moiety is -NH${}_{2}$, which allows us to form amides [12], diazo compound [22], and Schiff bases [13]. The transformation of amine moieties into amides was the pioneer path reported for covalent PSM [12,13]. For the amide formation, the carboxylate group needs to be activated, e.g., using an acyl chloride or anhydride. The atom economy when using an acyl chloride or non-cyclic anhydride is not favorable, as reagents excess, organic solvents, and bases should be included in the atom economic equation, and many of these chemicals require additional safety precautions, e.g., pyridine, acyl chloride. To our knowledge, there is only one report of amide direct reaction using formic acid and an amino moiety in the ligand [24]. Additionally, in our case, the atomic economy is highly favorable due to the generation of water as the only by-product.

In this manuscript, we report on the post synthetic modification of Al-BDC-NH${}_{2}$ metal organic framework, and its potential use as a fluoride sensor in water samples.

## 2. Materials and Methods

### 2.1. Sample Preparation

The synthesis of -NH${}_{2}$ tagged MOF was conducted according to a slightly modified procedure previously reported [25]. In a 250 mL glass bottle, aluminum (III) chloride hexahydrate (3.06 g) and 2-amino terephthalic acid (3.36 g) were dissolved in 180 mL N,N-dimethylformamide (DMF) at 80 °C and maintained at 130 °C for 24 h. The MOF was isolated by centrifugation and washed three times with DMF and ethanol. For thermal covalent PSM, 1.5 g of Al-BDC-NH${}_{2}$ was added to a 100 mL ethanol solution of each dye, dispersed in an ultrasonic bath and under agitation overnight in an orbital shaker. The suspension was decanted and the solid was dried in an oven at 140 °C overnight. The solids were dispersed in clean ethanol in an ultrasonic bath and filtered to remove excess dye. The process was repeated four times, followed by soaking in ethanol for 48 h to remove the remaining dye excess. The resulting clean solids were filtered and dried at 80 °C. For the SCN covalent PSM, Al-BDC-NH${}_{2}$ was modified with isothiocyanates according to a previously reported procedure [25]. A fluorescein 5(6)-isothiocyanate (30.0 mg) solution was prepared by dissolving it in absolute ethanol (100 mL). The same procedure was used for rhodamine B Isothiocyanate solution (42.0 mg). Al-BDC-NH${}_{2}$ (200 mg) were added to the ethanolic dye solutions and agitated for 48 h in an orbital shaker at room temperature. The isothiocyanate-modified MOFs were filtered and repeatedly washed with ethanol and deionized water. The resulting solids were dried at 80 °C.

### 2.2. Characterization

X-ray diffractograms (XRD) of powders were recorded on a Rigaku X-ray diffractometer RINT-Ultima/S2K (Cu Kα source). Textural properties were obtained from N${}_{2}$ adsorption data obtained at 77 K in a manometric adsorption equipment (Micromeritics, TriStar). For nuclear magnetic resonance (NMR) the samples were dissolved in D${}_{2}$O and NaF and recorded on a Bruker AVANCE 400 FT NMR spectrometer. For the spectrophotometric quantification of ligand substitution, around 10 mg MOF and dyes were dissolved in 1000 mg·L${}^{-1}$ NaF aqueous solutions. Dilutions to fit the concentration in the measuring range of the equipment were also performed with 1000 mg·L${}^{-1}$ NaF aqueous solutions. Absorbance measurements were performed on an Ocean Optics Spectrophotometer, emission measurements were performed on the same spectrometer using a LED for sample excitation (450 nm for fluorescein and 530 nm for the other dyes).

For the progressive etching of MOF particles, a standard suspension was prepared by adding 120 mg of the erythrosine B modified MOF to 1.2 mL of MilliQ water and dispersed in an ultrasonic bath for 20 min. Each measurement was done by adding 90 uL of the mentioned suspension to 5 mL of the fluoride solution, homogenized in an orbital shaker overnight, filtered, and diluted. Complete dissolution of the MOF was achieved using 1000 mg·L${}^{-1}$ fluoride solution. The selectivity of the MOF for fluoride sensing was performed using a 5 mg·L${}^{-1}$ of fluoride, sulfate, phosphate, acetate, chloride, nitrate, bromide, iodide, sodium, calcium and magnesium solutions in deionized water.

## 3. Results

### 3.1. Post-Synthetic Modification of Al-BDC-NH${}_{2}$ MOF

Thermal condensation of Al-BDC-NH${}_{2}$ MOF with xanthene dyes (Fluorescein, rhodamine B, eosin Y, erythrosine B and rose bengal) without using any previous activation yielded strongly colored solids, as shown in Figure 1a. Due to the complex signals in the carboxy region, the FTIR measurements (Figure 1b,c) were not conclusive to elucidate the formed bond. To determine the type of bond between the dye and MOF, all samples were dissolved and analyzed by ${}^{13}$C NMR. The signal at 162 ppm in the ${}^{13}$C NMR spectra indicates the formation of an amide bond, which is in agreement with the previously reported chemical shifts for this bond [26] (Figure 1d). Figure 1e shows the proposed synthetic pathway for the formation of the amide in the thermally activated PSM experiments.

### 3.2. Structural and Textural Characterization of PSM-MOFs

The structure of the MOF treated with different dyes was studied by XRD. Figure 2a shows the XRD of modified and unmodified MOFs, showing the same main peaks of the Al-BDC-NH${}_{2}$ between 5 and 12°, which agree with previous reports [27]. The peak broadening in the diffraction pattern indicates small crystalline domains. The PSM samples also exhibited less intensity in the peaks at higher angles with respect to the signals from the raw Al-BDC-NH${}_{2}$ sample. The decrease in peak intensity could be related to the degree of substitution and the presence of heavy atoms (Cl, Br, and I) in the dyes.

N${}_{2}$ isotherms and corresponding pore size distributions are shown in Figure 2b,c. Texture parameters are shown in Table 1. The S${}_{BET}$ for Al-BDC-NH${}_{2}$ agrees with values previously reported ( 1500 m${}^{2}$/g) [28]. All modified MOFs exhibited a reduction in the S${}_{BET}$ after PSM due to the introduction of organic moieties in the pores. The same tendency was observed in the total pore volume and micropore volume. Al-BDC-NH${}_{2}$ exhibits Type Ib isotherm, which transforms into a Type Ia when reducing the surface area. Type Ib isotherm corresponds to wider micropores, while Type Ia corresponds to narrow micropores. The transition in the isotherm type and changes in the surface area indicates a reduction in the pore size resulting from the introduction of dyes into the pores of the MOF. The substituted samples exhibit a hysteresis loop, H4 type, which indicates slit-shaped pores, which is in agreement with the presence of rigid flat xanthene-dyes in the pores [29,30]. Samples with lower surface area, those modified with rhodamine and rose bengal, showed a hysteresis below the p/p${}_{o}$ limit reported for N${}_{2}$ at 77K closure (p/p${}_{o}$∼0.42), indicating an irreversible change. The reason for this change is under consideration and will be part of more detailed work to be published in the future.

The incorporation of the dye was spectrophotometrically quantified, and results are shown in Figure 3. The dye incorporation quantification was performed using the same dye used for MOF modification as standards and assuming the absorption coefficient is not modified by the PSM. The reaction yield values varied between 0.4 and 3.5% (Figure 3a). The atom economy of the reaction is elevated due to water being the only by-product. Additionally, the moderate reaction temperature used and the thermal stability of the dyes [31,32,33] would allow the distillation of impregnation and washing solutions to recover the unreacted dye and ethanol.

The absorption and emission spectra of the modified MOFs dissolved in fluoride aqueous solution and dyes are shown in Figure 3b–f. In the case of fluorescein (Figure 3b), the absorption maxima is not modified, but modified ligands exhibited an increase in the absorbance near the UV region, which is higher in the thermally modified ligand. The thermally modified ligands exhibit a minimal red shift with respect to the dye. In the case of the rhodamine B modified samples (Figure 3c), the emission spectra remain unmodified only showing a small shoulder around 520 nm. The emission of these samples also exhibits a red shift for thermally modified samples. Eosin Y (Figure 3d) does not show significant modification in the absorption and emission spectra. In the case of erythrosine B (Figure 3e) and rose bengal (Figure 3f), the emission spectra did not change, but the absorption spectra have a clear blue shift. A similar shift in the absorption spectra was previously reported when rose bengal was attached to collagen [34] and when erythrosine B interacted with different surfactants [35,36].

### 3.3. Chemical Interaction of Modified MOFs with Fluoride

Figure 4 shows the light absorbance spectra of fluoride etched MOF modified with erythrosine B in solution. This experiment monitors the release of modified ligand to the solution, which can be decoupled into two processes, the -OH/-F exchange in the open metal sites [5] and the metal -ligand bond breakage. We propose that the ligand exchange at the open metal sites occurs during the first stage in the whole particle due to the porous nature of the MOF, while ligand release by metal–ligand bond breakage occurs gradually due to the intrinsic constrains of the MOF’s reticular structure. The origin of these open metal sites can be from defects in structure and uncoordinated Al atoms in the surface of the particles. This process can be interpreted as a surface-to-center etching of the MOF particles.

In Figure 5, the recovery % of the signal for fluoride 5 mg·L${}^{-1}$ in the presence of other ions in equal concentration is shown. The presence of common anions in drinking water like sulphates, nitrates, chlorides and cations like calcium and magnesium does not produce any decrease of the measured absorbance, and therefore, showing a good selectivity for fluorides. Carbonates and phosphates changed the pH of the solution to higher values than 7, and thus, produced an excess recovery % as the MOF dissolution by pH was coupled with the MOF dissolution with fluorides.

## 4. Discussion

### 4.1. Post Synthetic Modification of MOFs

The post synthetic modification of Al-BDC-NH${}_{2}$ with the xanthene dyes through thermal condensation, showed yields similar to the isothiocyanate methodology, but with the advantage of being achieved in few steps, lower number of reagents and solvents used (with the possibility of recovering the non-reacted dyes) and lower costs of functionalizing reagents. The color change of the white Al-BDC-NH${}_{2}$ to yellow, orange, red and magenta resulting solids evidenced the successful interaction of the MOF with the dyes. The modification mechanism via thermal condensation can be understood by considering the amide formation between the MOF’s ligand and the dye, with water as a by-product. Additionally, the high temperatures at which the heterogeneous condensation occurs allow the elimination of water, further increasing the reaction yield. The amide formation was confirmed by NMR ${}^{13}$C, and the main crystalline structure of the starting MOF was maintained and confirmed by powder XRD. The introduction of the xanthene dye to the MOF produced a decrease in the surface area and pore size.

### 4.2. Interpretation of the MOF-Fluoride Interaction

The MOF’s surface etching with fluorides found in Figure 4a initially shows a non-linear regime, which can be attributed to the ligand exchange (Equation (Equation 1)) on the open metal sites and partial metal–ligand bond breakage (Equations (Equation 2) and (Equation 4)) along with the release of the dye-modified (*L*${}_{B}$) and unmodified ligand (*L*${}_{A}$) to the solution (Equations (Equation 3) and (Equation 5)). This “mixed region” is constituted by absorbance “silent” (Equations (Equation 1)–(Equation 4)) and “non-silent” (Equation (Equation 5)) reactions. A linear region is also observed, where the ratio of fluoride addition to the dye-modified ligand is linear and can be mainly attributed to the process described in Equation (Equation 3). The last region is described by Equation (Equation 3), where a core with unmodified ligand released into the solution, exhibiting no changes in the absorbance. Additionally, it can be observed that the upper limit is asymptotically achieved, possibly due to the diffuse interface between the dye-modified shell and the unmodified core.
(1)$$∼Al-OH+F^{-}\to ∼Al-F+OH^{-}$$
(2)$$∼Al-L_{A}-Al\;+F^{-}+H_{2}O\to ∼Al-F+∼Al-L_{A}H+OH^{-}$$
(3)$$∼Al-L_{A}H+F^{-}+H_{2}O\to ∼Al-F+L_{A}H_{2}+OH^{-}$$
(4)$$∼Al-L_{B}-Al∼+F^{-}+H_{2}O\to ∼Al-F+∼Al-L_{B}H^{+}OH^{-}$$
(5)$$∼Al-L_{B}H+F^{-}+H_{2}O\to ∼Al-F+L_{B}H_{2}+OH^{-}$$

All the processes described above can be explained analogously to the hydrolysis and condensation phenomena described in the sol-gel process [37], where *L*${}_{A}$ corresponds to the unmodified ligand, *L*${}_{B}$ the modified one, and Al-X the metal centers in the MOF structure where X can be -OH, -F or L.

The strong interaction of fluoride ions with the open metal sites of the MOF and the ligand exchange proposed in Equations (Equation 1)–(Equation 5) conforms an interesting fluoride detection system, as the release of the dye due to hydrolysis can be monitored and correlated with the fluoride concentration present in water.

The analytical performance of the modified MOF in terms of sensitivity, interference, limit of detection, among other analytical parameters are out of scope for this work. However, due to the high selectivity of the modified MOF toward the presence of fluoride in water, the development of an analytical methodology for fluoride quantification will be further pursued. Additionally, the system shows a similar response to interference previously reported for other MOF based systems [5,25,38]. The presence of HCO${}_{3}^{-}$ and H${}_{2}$PO${}_{4}^{2-}$ increases the signal, overestimating the fluoride concentration due to the increase in pH in the sample.

## 5. Conclusions

A versatile covalent post-synthetic modification of MOFs was successfully developed allowing the introduction of complex moieties via thermal condensation. The reaction allows to perform PSM without activation and using mild reaction temperatures compatibles with the thermal stability of many MOFs and dyes. The synthetic path can be extended to more moieties considering the boiling point of the reagents, the relationship between the pore size of the MOFs and dynamic radius of the reagents and the presence of synthons of the desired function groups, -NH${}_{2}$ and -COOH for amides, -OH and -COOH for esters. The atom economy of the reported method is elevated due to the production of water as the only by-product and the possible recovery of the unreacted dyes due to moderate reaction temperatures. The reaction yield was between 0.4 and 3.5%, which were in the same order of magnitude as the previously reported isothiocyanate method. The proposed method has the advantage of having a lower cost of reagents and an easier storage condition for them, as isothiocyanates should always be kept refrigerated. The MOF particle progressive etching in the presence of fluoride indicates a dye diffusion through the pores toward the center of the particle during thermal condensation. The fluoride etching of the MOF enables an interesting fluoride detection system based on fluorescence detection of the released dye, with high selectivity toward fluoride, even in the presence of common ionic concomitants commonly present in water.

## Figures and Tables

**Figure 1 bioengineering-08-00196-f001:**
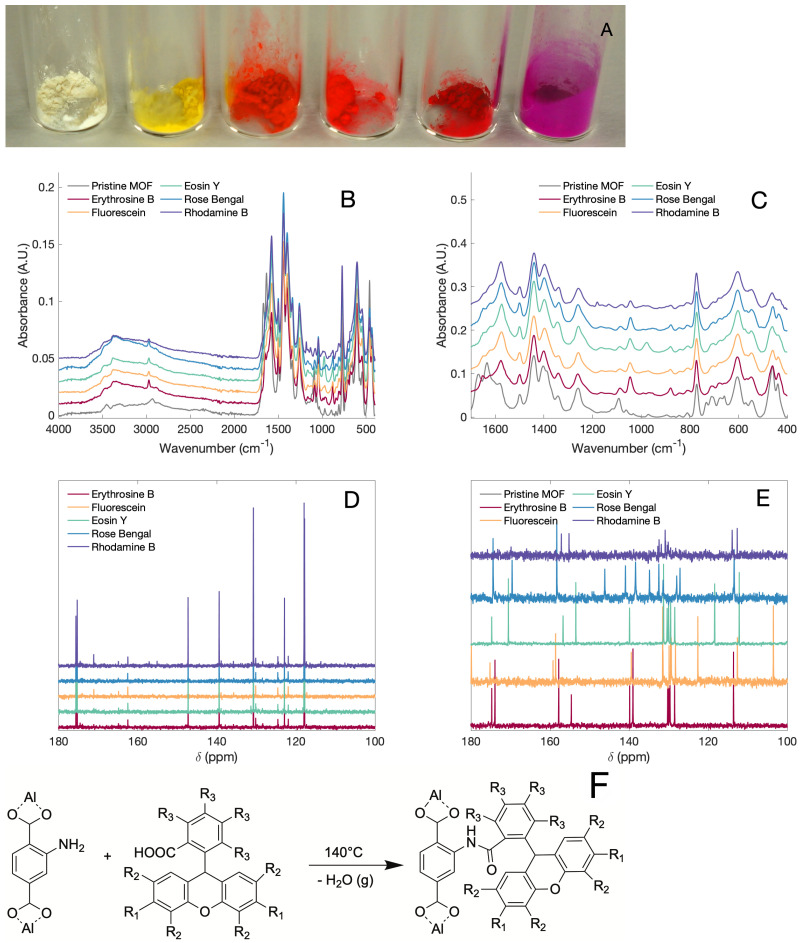
(**A**) Photograph of the modified MOF, from left to right: Al-BDC-NH${}_{2}$, fluorescein, eosin Y, erythrosine B, rose bengal and rhodamine B. FTIR spectra from the modified and unmodified MOFs, (**B**) Complete spectra and (**C**) carboxylate region. (**D**) ${}^{13}$C NMR of the modified MOF dissolved in D${}_{2}$O/NaF. (**E**) ${}^{13}$C NMR of the dyes used in the MOF modification (**F**) Proposed synthetic pathway for the formation of the amide. For fluorescein: R${}_{1}$ = -OH, R${}_{2}$ =R${}_{3}$ = -H, rhodamine B: R${}_{1}$ = -N(Et)${}_{2}$, R${}_{2}$ = R${}_{3}$ = -H, rose bengal: R${}_{1}$ = -OH, R${}_{2}$ = -I, R${}_{3}$ = -Cl, eosin Y: R${}_{1}$ = -OH, R${}_{2}$ = -Br, R${}_{3}$ = -H, and erythrosine B: R${}_{1}$ = -OH, R${}_{2}$ = -I, R${}_{3}$ = -H.

**Figure 2 bioengineering-08-00196-f002:**
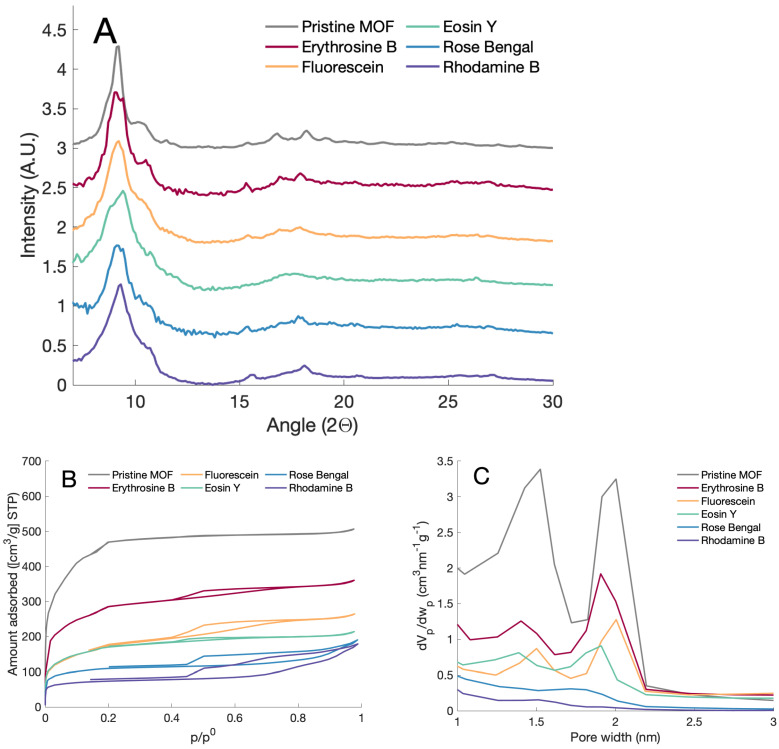
(**A**) X-ray diffraction patterns from the modified and unmodified MOFs. (**B**) Adsorption–desorption isotherms of nitrogen at 77 K of the modified and unmodified MOFs. (**C**) Pore size distribution obtained from the isotherms measurements.

**Figure 3 bioengineering-08-00196-f003:**
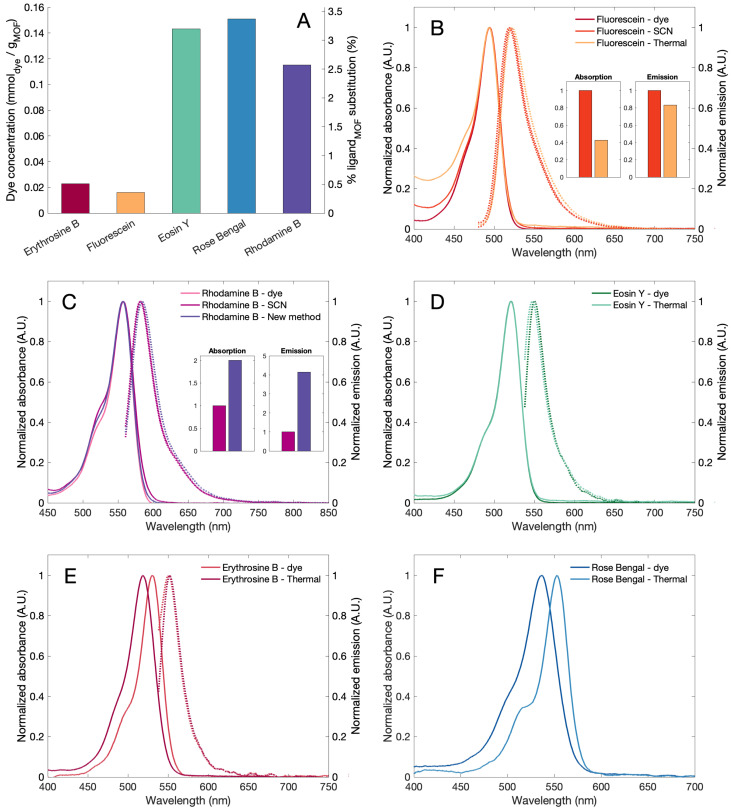
(**A**) Quantification of dye loading by the proposed method, left scale is dye quantification assuming similar absorption coefficient for dye and modified ligand, right scale shows the approximately ligand substitution in the MOF assuming a Mr${}_{MOF}$ = 223 g mol${}^{-1}$. (**B**) Normalized absorption and emission spectra of fluorescein and fluorescein modified MOFs. Inset shows the comparison of absorption and emission of fluorescein-modified ligands by both methods. (**C**) For rhodamine B and rhodamine B modified MOFs. (**D**) For Eosyn Y and Eosyn Y modified MOFs. (**E**) For Erythrosine B and Erythrosine B modified MOFs. (**F**) For Rose Bengal and Rose Bengal modified MOFs. Note: dye corresponds to the dye in solution, -SCN for the isothiocyanate modified ligand in solution and—thermal for the proposed method modified ligand in solution.

**Figure 4 bioengineering-08-00196-f004:**
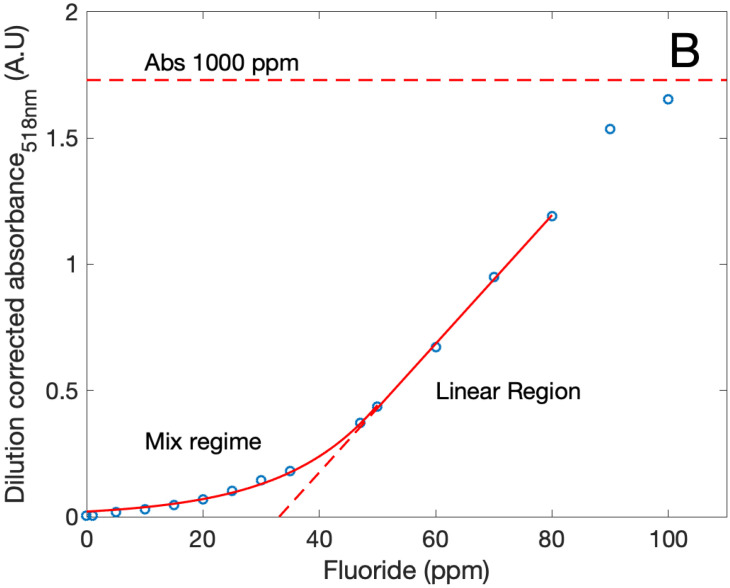
Absorbance of the solution in function of fluoride concentration when erythrosine B modified MOFs is etched with fluorides.

**Figure 5 bioengineering-08-00196-f005:**
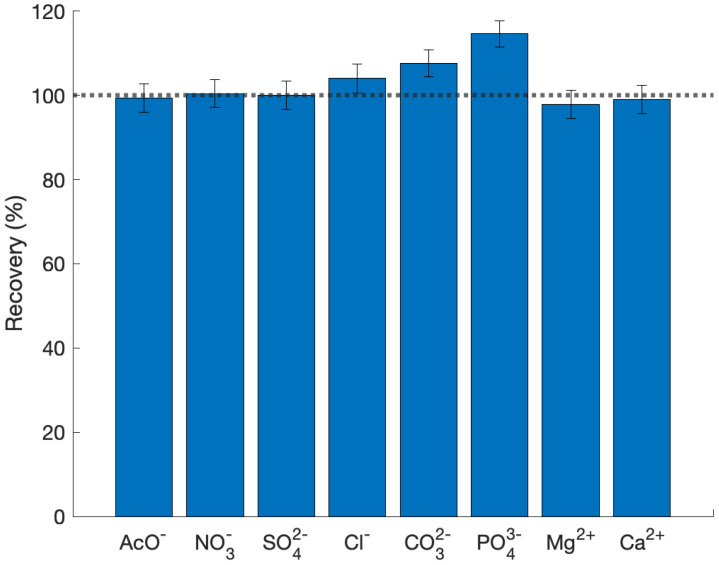
Sensitivity of the modified MOF (Al-BDC-Rodhamine B) in the presence of fluoride and other concomitant ions.

**Table 1 bioengineering-08-00196-t001:** BET areas obtained from the isotherms for pristine Al-BDC-NH${}_{2}$ and modified MOF with dyes.

Sample	*S*${}_{\mathrm{BET}}$ (m${}^{2}$/g)	V${}_{\mu }$P (cm${}^{3}$/g)	V${}_{\mathrm{TP}}$ (cm${}^{3}$/g)
Pristine MOF	1386	0.599	0.784
Fluorescein	559	0.221	0.409
Rhodamine B	222	0.111	0.277
Rose bengal	322	0.161	0.295
Eosin Y	521	0.225	0.331
Erythrosine B	864	0.394	0.558

## Data Availability

Not applicable.

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
