# Peer review of "Versatile Covalent Postsynthetic Modification of Metal Organic Frameworks via Thermal Condensation for Fluoride Sensing in Waters"

_bioengineering, 2021, doi:10.3390/bioengineering8120196_

Round 1
Reviewer 1 Report
In general, this manuscript reports the postsynthetic modification of MOFs and their application in fluoride sensing. From synthetic viewpoint, the proposed thermal condensation is relatively easy with low by-products, and the sensing is also performed. Overall, this manuscript is basically well-organized, and the idea is interesting. However, there are some problems and thus I suggest publishing the manuscript with a major revision.
- There are some typos, singular and plural, and grammatical mistakes in the manuscript, such as: “science cross their ways1”, “Texture parameters are shown in Table ??.”, “exchange in the open 167 metal sites27 and”, “these open metal sites can be from defects”, just to name a few.
- When discussing the post-synthetic modifications, similar thermal condensation modifications in porous frameworks are suggested to be cited, like Nat. Commun., 2018, 9, 3933, and https://doi.org/10.1039/C8SC04518F;
- In the introduction part, too much discussion on the PSM, whereas there is little progress on the fluoride sensor in drinking water. Please give a balance and highlight the unique property using the present PSM.
Author Response
- There are some typos, singular and plural, and grammatical mistakes in the manuscript, such as: “science cross their ways1”, “Texture parameters are shown in Table ??.”, “exchange in the open 167 metal sites27 and”, “these open metal sites can be from defects”, just to name a few.
We apologize for the mistakes; we have carefully proofread the manuscript and corrected the latex file.
-When discussing the post-synthetic modifications, similar thermal condensation modifications in porous frameworks are suggested to be cited, like Nat. Commun., 2018, 9, 3933, and https://doi.org/10.1039/C8SC04518F;
We thank the reviewer for the suggestion to improve the manuscript quality. We included the suggested cites.
- In the introduction part, too much discussion on the PSM, whereas there is little progress on the fluoride sensor in drinking water. Please give a balance and highlight the unique property using the present PSM.
We improved the balance between the PSM and fluoride sensing in the introduction, 30% fluoride problem, 30% introduction to MOFs and 40% MOFs’ PSM. We appreciate the observation to improve the quality of the manuscript.
Reviewer 2 Report
Authors reported on a post synthetic functionalization of MOFs for the sensing of fluoride ions in water. The thermal and thiocyanate method of functionalization have been compared. Modifications were conducted using common xanthene dyes. This work can be published after considering some issues listed below:
- To prove the formation of new amide bond between MOF and dye the C13 NMR spectra and PXRD patterns of substrates should be compared with products.
- The caption for Figure 1 should be corrected.
- The research should be improved by reusability study.
- LOD and LDR parameters should be determined for this method of fluorine detection.
- Whether the tests were carried out on real samples? It would enrich this work.
- It would be nice to compare this method with other reports of fluoride sensors.
Author Response
-To prove the formation of new amide bond between MOF and dye the C13 NMR spectra and PXRD patterns of substrates should be compared with products.
According to reviewer’s comment, we corrected the XRD pattern to make it more clear and included the 13C NMR spectra of the dyes.
- The caption for Figure 1 should be corrected.
We apologize for the mistake; the caption was corrected.
-The research should be improved by reusability study.
We thank for the suggestion. The MOFs cannot be reused for several fluoride determinations. But our preliminary test suggests that the required amount of MOF for a determination is around 1mg, which makes the costs affordable for rural communities without access to safe water and cheaper than commercial kits.
-LOD and LDR parameters should be determined for this method of fluorine detection.
We agree that limit of detection and linear range should be included but the system has a complex physical chemistry and the implementation in an analytical methodology and ideally in a portable sensor requires more experimental work. Figure 4 shows the absorbance of a solution in contact with MOF, it can be observed that there is an induction initial part which makes necessary to find the ideal ratio of MOF to sample to fit the linear range and detection limit in the requirements from WHO. All the required information to transform the system into a sensor is under development and will be included in a new manuscript showing the relevance of the materials for developing a fluoride sensor.
- Whether the tests were carried out on real samples? It would enrich this work.
We agree with this suggestion. To make a reliable comparison, natural waters should be characterized with many parameters: pH, conductivity, anionic composition, etc. Also, the method should be validated to a standardized methodology, like ionic chromatography. Due to the short times involved in this stage of revision, it is impossible to provide reliable information in this point. We thank for the suggestion, and we will include it in a future manuscript including the analytical chemistry aspects of this material.
We also clarified in Materials and Methods that the interference study was performed in deionized water.
- It would be nice to compare this method with other reports of fluoride sensors.
According to reviewers’ suggestions, we included references to reported MOFs based sensors. We thank for the suggestion to improve the manuscript quality.
Round 2
Reviewer 1 Report
It can be accepted in the present form
Reviewer 2 Report
The revised manuscript can be accepted by the “Bioengineering”.